# Quantum Perceptron Revisited: Computational-Statistical Tradeoffs

**Mathieu Roget**[1,2]  **Giuseppe Di Molfetta**[1]  **Hachem Kadri**[1]

[1]Aix-Marseille University, CNRS, LIS, Marseille, France
[2]École Normale Superieure de Lyon, Lyon, France

## Abstract

Quantum machine learning algorithms could provide significant speed-ups over their classical counterparts; however, whether they could also achieve good generalization remains unclear. Recently, two quantum perceptron models which give a quadratic improvement over the classical perceptron algorithm using Grover's search have been proposed by Wiebe et al. [2016]. While the first model reduces the complexity with respect to the size of the training set, the second one improves the bound on the number of mistakes made by the perceptron. In this paper, we introduce a hybrid quantum-classical perceptron algorithm with lower complexity and better generalization ability than the classical perceptron. We show a quadratic improvement over the classical perceptron in both the number of samples and the margin of the data. We derive a bound on the expected error of the hypothesis returned by our algorithm, which compares favorably to the one obtained with the classical online perceptron. We use numerical experiments to illustrate the trade-off between computational complexity and statistical accuracy in quantum perceptron learning and discuss some of the key practical issues surrounding the implementation of quantum perceptron models into near-term quantum devices, whose practical implementation represents a serious challenge due to inherent noise. However, the potential benefits make correcting this worthwhile.

## 1 INTRODUCTION

Quantum machine learning is an attractive field of research that contributes towards addressing the need for computationally efficient machine learning (ML) algorithms capable of handling huge amounts of data [Wittek, 2014, Biamonte et al., 2017, Ciliberto et al., 2018, Schuld and Petruccione, 2018, Dunjko and Wittek, 2020]. Previous works in the field have mainly investigated machine learning tasks when a quantum information processing device is used, showing that a significant speed-up can be achieved compared to classical ML algorithms [Rebentrost et al., 2014, Kerenidis et al., 2019, 2020, Arunachalam and Maity, 2020, Ma and Tresp, 2021]. Quantum computing promises the ability to solve intractable ML problems by harnessing quantum effects such as superposition and entanglement.

Quantum superposition, a fundamental concept in quantum computing, is the means by which quantum algorithms like Grover's search can outperform classical ones. Ordinary computers operate with states built from a finite number of bits. Each bit may exist in one of the two states, 0 or 1. A quantum computer works with a finite set of objects called qubits. Each qubit has two separate states, also denoted by 0 and 1, but a qubit can also be in what is called a "quantum superposition" of these states, in which it is, in some sense, both 0 and 1 simultaneously [Nielsen and Chuang, 2002]. Grover's algorithm is one of the most famous algorithm in quantum computing [Grover, 1996, Roget et al., 2020]. It solves the problem of finding one item from an unstructured database of $N$ items in time $O(\sqrt{N})$, so beating the classical $O(N)$ time requirement. Recent works have investigated the use of Grover's search algorithm to enhance machine learning and have proved its ability of providing computational speed-up over classical ML algorithms [Aïmeur et al., 2013, Wittek, 2014, Wiebe et al., 2016, Li et al., 2019, Casalé et al., 2020]. Beyond Grover's algorithm, quantum algorithms for linear algebra, such as quantum matrix inversion and quantum singular value decomposition, were developed and used in the context of machine learning [Rebentrost et al., 2014, Kerenidis and Prakash, 2017]. Among the quantum-enhanced ML algorithms that were proposed in the literature, quantum perceptron models in particular attracted our attention because it has been shown that they could enable non-trivial improvements not only in the computational complexity but also in

*Accepted for the 38th Conference on Uncertainty in Artificial Intelligence* (UAI 2022).

Table 1: Summary of the computational complexities and the expected risk bounds of the classical online perceptron and the quantum perceptron models.

| Algorithm | Complexity | Expected risk |
|---|---|---|
| CLASSICAL ONLINE PERCEPTRON [e.g., Mohri et al., 2018] | $O\left(\dfrac{N}{\gamma^2}\right)$ | $\leq \underset{S\sim\mathcal{D}^{N+1}}{\mathbb{E}}\left(\dfrac{\min(M(S),\frac{1}{\gamma_S^2})}{N+1}\right)$ |
| ONLINE QUANTUM PERCEPTRON [Wiebe et al., 2016] | $O\left(\dfrac{\sqrt{N}}{\gamma^2}\ln\left(\dfrac{1}{\epsilon\gamma^2}\right)\right)$ | n/a |
| VERSION SPACE QUANTUM PERCEPTRON [Wiebe et al., 2016] | $O\left(\dfrac{N}{\sqrt{\gamma}}\ln^{3/2}1/\epsilon\right)$ | n/a |
| HYBRID QUANTUM PERCEPTRON (this work) | $O\left(\dfrac{\sqrt{N}}{\gamma}\ln(1/\epsilon)\ln\left(\dfrac{1}{\gamma\epsilon}\right)\right)$ | $\leq \underset{S\sim\mathcal{D}^{N+1}}{\mathbb{E}}\left(\sqrt{\dfrac{\pi}{2}}\dfrac{\ln 1/\epsilon}{N+1}\dfrac{1}{\gamma_S}\right)$ |

the statistical performance of the perceptron [Wiebe et al., 2016]. This may support the (beneficial) effect of quantum computations on generalization performance.

In Wiebe et al. [2016], two quantum perceptron models based on Grover's search algorithm were introduced. The first one (namely ONLINE QUANTUM PERCEPTRON) is a quantum extension of the classical online perception algorithm. The complexity of the online quantum perceptron with respect to the number of examples $N$ is $O(\sqrt{N})$, which is a quadratic improvement over the classical perceptron. However, no improvement in the number of updates made by the perceptron was achieved, as its mistake bound is $O(1/\gamma^2)$, where $\gamma$ is the margin, which is the same as in the classical case. The second quantum perceptron model (namely VERSION SPACE QUANTUM PERCEPTRON) is based on the notion of version space [Herbrich et al., 2001, Mitchell, 1982] and has a mistake bound of $O(1/\sqrt{\gamma})$, which is a substantial improvement over the classical online perceptron. Yet, as with the classical perceptron, the computational complexity of the algorithm is linear in $N$. Hence, the question arises whether it is possible to design quantum algorithms for perceptron learning that enjoy the best features of both types of quantum perceptron models. In other words, can we develop a quantum perceptron algorithm that provides improvements in both the computational complexity and the number of mistakes the perceptron makes?

This paper provides, to the best of our knowledge, the first perceptron learning algorithm that has lower complexity and better generalization ability than the well-known classical online perceptron. Specifically, we make the following contributions: i) we introduce a hybrid quantum-classical perceptron algorithm (namely HYBRID QUANTUM PERCEP-TRON) that performs a quantum search over the training set for randomly generated linear separators in order to find one that lies in the version space; ii) we show a quadratic

improvement over the classical perceptron in both the number of samples and the margin of the data; iii) we derive a bound on the expected error of the hypothesis returned by our algorithm that compares favorably to the one obtained with the classical online perceptron; iv) we use numerical experiments to illustrate the trade-off between computational complexity and statistical accuracy in quantum perceptron learning and discuss some of the key practical issues surrounding the implementation of quantum perceptron models into near-term quantum devices, whose practical implementation represents a serious challenge due to inherent noise. Our theoretical results for Quantum Perceptron and other related works are summarized in Table 1.

## 2  PRELIMINARIES

We begin with reviewing the classical perceptron algorithm and then give some background on quantum computing and Grover's search algorithm.

### 2.1  CLASSICAL PERCEPTRON ALGORITHM

**Algorithm and complexity**   The perceptron is an on-line algorithm designed to solve binary classification problems [Rosenblatt, 1958]. It has received a lot of attention due to its simplicity and versatility [Cesa-Bianchi et al., 2005, Freund and Schapire, 1999, Shalev-Shwartz and Singer, 2005]. Consider a training set $\{(x_1,y_1),...,(x_N,y_N)\}$ with data vectors $x_i \in \mathbb{R}^D$ and class labels $y_i \in \{-1,1\}$, $i = 1,\ldots,N$. We assume that the data are linearly separable, i.e., there exists a hyperplane that separates the data points of the class 1 from those of the class $-1$. The CLAS-SICAL ONLINE PERCEPTRON will find a separator $w \in \mathbb{R}^D$ such that $\forall i,\ y_iw^Tx_i \geq 0$. The algorithm simply updates the vector $w$ each times it misclassifies a point. The CLAS-

SICAL ONLINE PERCEPTRON is depicted in Algorithm 1.

The margin $\gamma$ between the two classes is defined by:

$$\gamma = \max_{v \in \mathbb{R}^D} \min_{1 \leq i \leq N} \frac{y_i \langle v, x_i \rangle}{\|v\|}.$$

Usually, the margin is small (close to zero) which means that the classes are close and separating them is hard. In the following, we always assume that the margin is smaller than one (which can be achieved by normalizing the training set) and the asymptotic complexities are studied when $N$ and $\frac{1}{\gamma}$ are large. When the norm of the $x_i$'s is at most 1, it holds that the number of updates made by the perceptron during the learning phase is at most $O(\frac{1}{\gamma^2})$. This result is known as the bound of Novikoff [Novikoff, 1962, Mohri and Rostamizadeh, 2013]. If we want to correctly classify all the $N$ samples, the final complexity of the perceptron is then $O(\frac{N}{\gamma^2})$.

**Generalization** One of the most fundamental questions in Machine Learning is what are the generalization guarantees of a learning algorithm. The perceptron algorithm learns a mapping between input data and target labels using a finite sample of labeled examples, and then uses a hyperplane to separate the data and predict the class of unseen examples. It is therefore important to assess the ability of the perceptron to generalize to unseen data. In a statistical learning framework, such assessment is often performed by bounding the risk (or generalization error). Let us denote by $\mathcal{D}$ the distribution that generates the data. The training sample $S$ of $N$ data points $\{(x_i, y_i)_{i=1}^N\}$ is assumed to be drawn randomly from the (unknown) distribution $\mathcal{D}$ and we write $S \sim \mathcal{D}^N$. The binary classification risk is defined by

$$R(h_S) = \mathbb{E}_{(x,y) \sim \mathcal{D}} (\mathbb{1}\{h_S(x) \neq y\}) ,$$

where $h_S$ is the hypothesis returned by the algorithm on the sample $S$.

**Theorem 1 .** *Assume that the data are linearly separable. Let $h_S$ be the hypothesis returned by the* CLASSICAL ONLINE PERCEPTRON *algorithm after training over a sample $S$ of size $N$ drawn according to some distribution $\mathcal{D}$. We note $\gamma_S$ the margin of sample $S$. Then, the expected risk of $h_S$ is bounded as follows:*

$$\mathbb{E}_{S \sim \mathcal{D}^N} (R(h_S)) \leq \frac{1}{N+1} \mathbb{E}_{S \sim \mathcal{D}^{N+1}} \left( \min(M(S), 1/\gamma_S^2) \right),$$

*where $M(S)$ is the number of updates made by the algorithm after training over $S$.*

*Proof.* See [Mohri et al., 2018, Theorem 8.9]. $\square$

---

**Algorithm 1:** CLASSICAL ONLINE PERCEPTRON

**Input:** data $(x_i, y_i)_{1 \leq i \leq N}$ ;                    // training set
$w \leftarrow 0$ ;                                       // separator in $\mathbb{R}^D$
**while** $(x_t, y_t) \leftarrow$ RECEIVE() ;        // data we receive
**do**
     **if** $y_t w^T x_t \leq 0$;        // data wrongly classified
     **then**
        $\lfloor$ $w \leftarrow w + y_t x_t$ ;   // we update the separator
**return** $w$

---

## 2.2 QUANTUM COMPUTATION AND GROVER'S SEARCH ALGORITHM

Before we introduce the quantum perceptron algorithm, we believe it is opportune to briefly present the principles of quantum mechanics, i.e. the underlying mathematical structure of all quantum physical systems. It is not possible to provide here a complete and exhaustive presentation, so we will limit ourselves to introduce only those "game rules" useful to understand the content of the following algorihtms, leaving it to the reader's curiosity to more complete reviews, such as Nielsen and Chuang [2002]. At best, we first introduce the arena where the game goes on, then we define the dynamics of the quantum system and finally we shortly introduce the measurement operation.

While classically a computational state takes value in $\{0, 1\}$, a quantum state is represented by a unit complex vector $|\psi\rangle$ in the Hilbert space $\mathbb{C}^2$. Such state space is equipped by an orthonormal basis $\{|0\rangle, |1\rangle\}$, such that any vector is generally described by a convex linear combination

$$|\psi\rangle = \alpha_0 |0\rangle + \alpha_1 |1\rangle ,$$

where $(\alpha_i)_{i=0,1}$ are complex numbers. More in general qubit basis states can also be combined to form product basis states to describe multi-qubits systems. If $|\psi_1\rangle, |\psi_2\rangle, ..., |\psi_n\rangle$ represent the states of $n$ isolated quantum systems, the state of the composite system is given by the tensor product of the state space of the components : $|\psi_1\rangle \otimes |\psi_2\rangle \otimes ... \otimes |\psi_n\rangle$. A concrete example of composite system is the memory of a $n-$qubit quantum computer, where each qubit is called register. In that case,

$$|\psi\rangle = \sum_{i=0}^{2^n-1} \alpha_i |i\rangle.$$

Similarly to classical computing, we can act by means of logical gates onto such quantum register to perform computation. Quantum circuits are nothing but reversible logical circuits onto complex-valued state space. Each quantum gate requires a special kind of reversible function, namely a unitary mapping, that is, a linear transformation of a complex inner product space that preserves the Hermitian inner product. When such systems are kept isolated, the

---

**Algorithm 2:** QSearch

**Input:** data $\{x_i\}_{1 \leq i \leq N}$ ; // data we want to search in

**Input:** oracle $f$ ; // oracle such that $f(x_i) = \mathbb{1}\{i \in \mathcal{M}\}$

$\psi_0 \leftarrow \dfrac{1}{\sqrt{N}} \displaystyle\sum_{i=1}^{N} |i\rangle$

$R \leftarrow \text{QUANTIFY}(f)$ // quantum version of the oracle

$U_g \leftarrow GR$

$m \leftarrow \mathcal{U}\left(\left\{0, \ldots, \left\lceil \dfrac{1}{\sin(2\sin^{-1}\left(\sqrt{\frac{1}{N}}\right))} \right\rceil - 1\right\}\right)$

$v \xleftarrow{\text{Meas}} U_g^m \psi_0$

**return** $v$

---

computation is kept reversible. However we need to obtain classical information about the outcome of a quantum computation task. In practice a quantum state has to be measured which formally coincides with an orthogonal projector onto one of the computational basis state $|v\rangle \in \mathbb{C}^n$. During such measurement operation, the quantum state is randomly collapsed into a classical state, with probability $\mathbb{P}\left(v = i \mid v \xleftarrow{\text{Meas}} \psi\right) = |\alpha_i|^2$, $\forall\, 0 \leq i < 2^n$, where $v$ has been expressed in a decimal system.

At the heart of the quantum perceptron algorithm lies the quantum search algorithm, which is widely used as main routine in many algorithms, generally guaranteeing to speed up any brute force $O(N)$ problem into a $O(\sqrt{N})$ problem. It has been introduced by Grover [1996] as a fast quantum mechanical algorithm for database search algorithm and it represents one of the most important and studied algorithm in quantum computing. In the following, we shortly present the Grover algorithm. Let us consider $N = 2^n$ elements and $\mathcal{M} \subseteq \{1, \ldots, N\}$ the searched elements. We start with the diagonal quantum state

$$|\psi_0\rangle = \frac{1}{\sqrt{N}} \sum_{i=1}^{N} |i\rangle \ .$$

We then apply two operators: an oracle and a reflection. The oracle $R$ is defined by

$$R |x\rangle = \begin{cases} -|x\rangle & \text{if } x \in \mathcal{M} \\ |x\rangle & \text{otherwise,} \end{cases}$$

while the reflection $G$ is given by

$$G = 2\psi\psi^\dagger - \mathbb{1} \ .$$

This two operators can in fact be view in a geometric way. We note $\#\mathcal{M}$ the cardinal of $\mathcal{M}$. Let's denote $a = \frac{\#\mathcal{M}}{N}$ the probability to find a searched element before running the algorithm (when the state is diagonal) and $\theta_a = \sin^{-1}(\sqrt{a})$, the angle between the subspace composed by the searched elements and the complementary subspace. Then one can

show that $U_g := GR$ is a rotation of an angle $2\theta_a$, meaning that after $j$ steps the probability to measure a searched element is

$$\mathbb{P}\left(v \in \mathcal{M} \mid v \xleftarrow{\text{Meas}} U_g^j \psi_0\right) = \sin^2\left((2j+1)\theta_a\right) \ .$$

It is then easy to find the number of steps that gives the optimal probability of finding a searched element. But to find this optimal number of steps, one needs to know $\theta_a$ which is directly related to the number of searched elements. We want here to adapt this algorithm in order to make it for an unknown number of searched elements.

The idea here that comes from Boyer et al. [1998] is simply to draw the number of steps randomly uniformly between 0 and $M - 1$. The resulting probability is

$$\mathbb{P}\left(v \in \mathcal{M} \mid v \xleftarrow{\text{Meas}} U_g^m \psi_0, m \leftarrow \mathcal{U}_{\{0, \ldots, M-1\}}\right)$$
$$= \frac{1}{M} \sum_{j=0}^{M-1} \sin^2\left((2j+1)\theta_a\right) = \frac{1}{2}\left(1 - \frac{\sin(4M\theta_a)}{2M\sin(2\theta_a)}\right) \ .$$

If $M \geq \frac{1}{\sin(2\theta_a)}$, then it holds that this probability is at least $\frac{1}{4}$. The last thing we need is to express a bound for $M$ that doesn't depend on $\theta_a$:

$$M \geq \frac{1}{\sin(2\theta_a)} = \frac{1}{\sin(2\sin^{-1}\left(\sqrt{\frac{\#\mathcal{M}}{N}}\right))}$$
$$\leq \frac{1}{\sin(2\sin^{-1}\left(\sqrt{\frac{1}{N}}\right))} = O(\sqrt{N}) \ .$$

In other words, we bound $M$ by its maximum value which occurs when $\#\mathcal{M} = 1$ (i.e. one marked element). The detailed quantum search over an unknown number of searched elements is given in Algorithm 2. This algorithm find a searched element with probability at least $\frac{1}{4}$ and has a complexity $O(\sqrt{N})$. By repeating the algorithm a logarithmic number of times, we can increase the probability of success to $1 - \epsilon$ for any $\epsilon > 0$ [Wiebe et al., 2016].

# 3 EXISTING QUANTUM PERCEPTRON ALGORITHMS

In this section, we discuss two existing quantum perceptron algorithms proposed in Wiebe et al. [2016] that are closely related to our work. Note that other quantum perceptron models can be found in the literature of quantum neural networks [Behrman et al., 2000, Ricks and Ventura, 2003, Schuld et al., 2015].

## 3.1 ONLINE QUANTUM PERCEPTRON

The classical online Perceptron updates the hyperplane when an example is misclassified and stops when all training

**Algorithm 3:** ONLINE QUANTUM PERCEPTRON [Wiebe et al., 2016]

**Input:** data $(x_i, y_i)_{1 \leq i \leq N}$ ;  `// training set`
$w \leftarrow 0$ ;  `// separator in $\mathbb{R}^D$`
**for** $i \in \{1, \ldots, 1/\gamma^2\}$ ;  `// we perform enough updates`
**do**
    **for** $j \in \left\{1, \ldots, \left\lceil \log_{3/4}(\gamma^2 \epsilon) \right\rceil \right\}$ ;  `// we increase the probability of QSEARCH`
    **do**
        $m \leftarrow \text{QSEARCH}(\{(x_k, y_k)\}_k)$ ;  `// searching for a point $x_m$ misclassified by $w$`
        **if** $y_m w_i^T x_m \leq 0$ ;  `// If actually misclassified...`
        **then**
            $w \leftarrow w + y_m x_m$ ;  `// ... then update`

**return** $w$

---

**Algorithm 4:** VERSION SPACE QUANTUM PERCEPTRON [Wiebe et al., 2016]

**Input:** data $(x_i, y_i)_{1 \leq i \leq N}$ ;  `// training set`
Draw $\{w_1, \ldots, w_K\} \leftarrow \mathcal{N}(0, \mathbb{1})$ ;  `// We assume this is done efficiently`
**for** $i \in \left\{1, \ldots, \left\lceil \log_{3/4}(\epsilon) \right\rceil \right\}$ ;  `// we increase the probability of QSEARCH`
**do**
    $m \leftarrow \text{QSEARCH}(\{w_k\}_k)$ ;  `// searching for a separator $w_m$ that correctly classifies the data`
    **if** $y_j w_m^T x_j > 0, \ \forall j$ ;  `// check if the obtained hyperplane is a good one`
    **then**
        **return** $w_m$

**return** $w_1$

---

data are correctly classified. The online quantum perceptron works similarly to the classical one. The main difference is the means by which misclassified points are detected. Instead of testing each point one by one, a Grover search is performed to find a wrongly classified example. Once this is done, the hyperplane is updated and the process is repeated until convergence. the ONLINE QUANTUM PERCEPTRON is outlined in Algorithm 3. Note that this algorithm is not really an online algorithm since it considers a quantum superposition of states representing the training data samples. The naming 'online' quantum perceptron is used because this algorithm has the same update rule than the classical online perceptron. In this quantum version of the perceptron, the computational complexity is improved from $O(N)$ to $O(\sqrt{N})$ due to the Grover search. However, an additional $\log\left(1/(\epsilon\gamma^2)\right)$ will appear to deal with the probability of failure of the quantum search. This is summarized in the theorem below.

**Theorem 2** [Wiebe et al., 2016]. *Let $S$ be a linearly separable sample of $N$ points of margin $\gamma$. Algorithm* ONLINE QUANTUM PERCEPTRON *finds a perfect separator with probability at least $1 - \epsilon$ and has a complexity of*

$$O\left(\frac{\sqrt{N}}{\gamma^2} \log\left(\frac{1}{\epsilon\gamma^2}\right)\right) .$$

## 3.2 VERSION SPACE QUANTUM PERCEPTRON

The idea of the second quantum perceptron model is based on the notion of version space, which is the set of hypotheses that are consistent with the training data [Herbrich et al., 2001]. Here, $K$ linear separators are randomly drawn from the normal distribution $\mathcal{N}(0, \mathbb{1})$, so the problem becomes how to find one of these separators that is in the version space, i.e., correctly separates the data. Using a version space point of view, the perceptron learning problem is

transformed into a search problem and then quantum search algorithms can be used to solve it efficiently. The Grover search is now applied over the generated hyperplanes and not the training set as in the previous algorithm (see Algorithm 4). A significant improvement on the number of hyperplanes $K$ is achieved; however, a full pass over the training examples is needed to find the hyperplane that belongs to the version space. The computational complexity of the algorithm is $O(N\sqrt{K})$ with an additional $\log 1/\epsilon$ because of the probability of failure, as summarized in the theorem below. Note that Wiebe et al. [2016] provided a result about the number of hyperplanes that must be generated to guarantee that at least one of them is in the version space. Interestingly, this number depends on the margin of the data. Indeed, it was shown that the number of hyperplanes to be sampled is $K = O\left(\frac{\ln(1/\epsilon)}{\gamma}\right)$.

**Theorem 3** [Wiebe et al., 2016]. *Let $S$ be a linearly separable sample of $N$ points of margin $\gamma$. Algorithm* VERSION SPACE QUANTUM PERCEPTRON *finds a perfect separator with probability at least $1 - \epsilon$ and has a complexity of*

$$O\left(\frac{N}{\sqrt{\gamma}} \log^{3/2}\left(\frac{1}{\epsilon}\right)\right) .$$

As we can see, this algorithm does not improve the complexity with respect to the number of the training data $N$; but it has a better statistical guarantee than the classical perceptron, since the classical mistake bound of $O(1/\gamma^2)$ can be improved to $O(1/\sqrt{\gamma})$. In the next section we propose a quantum perceptron algorithm that has the two advantages of the online and the version space quantum perceptron: it provides improvements in both the computational complexity and the number of mistakes.

# 4 HYBRID QUANTUM PERCEPTRON: AN IMPROVED PERCEPTRON LEARNING

This section presents our main results. We introduce a hybrid quantum perceptron algorithm to take advantage of the two quantum perceptron models described above. We show a quadratic improvement over the classical perceptron in both the number of samples and the margin of the data. Then, we derive a bound on the expected error of the hypothesis returned by our algorithm.

## 4.1 ALGORITHM

The idea is also to draw randomly several linear separators following the normal distribution $\mathcal{N}(0, \mathbb{1})$ and then search for one in the version space, so it correctly separates the data. However, in contrast to the VERSION SPACE QUANTUM PERCEPTRON, our algorithm will perform a quantum search over the training set for each separator to find a solution, and not a quantum search over the separators. By doing this, we can improve the complexity with respect to the number of samples $N$, as for the ONLINE QUANTUM PERCEPTRON, while still enjoying the benefits of the version space approach. Our hybrid quantum perceptron algorithm is described in Algorithm 5.

**Theorem 4 .** *Let $S$ be a linearly separable sample of $N$ points of margin $\gamma$. Algorithm* HYBRID QUANTUM PERCEPTRON *finds a perfect separator with probability at least $1 - \epsilon$ and has a complexity of*

$$
O\left(\frac{\sqrt{N}}{\gamma} \ln(1/\epsilon) \ln\left(\frac{1}{\gamma\epsilon}\right)\right) .
$$

*Proof.* See supplementary materials. ☐

This is a quadratic improvement in the computational and statistical complexity of the classical online perceptron. The improvement of the statistical complexity is quadratic only if we assume that the data supplied to the classical perceptron are provided the same way that the quantum one. Indeed, the complexity of the classical perceptron in this case is $O(\frac{N}{\gamma^2} \log(\frac{1}{\epsilon\gamma^2}))$ (see Wiebe et al. [2016, Th. 1]). If the classical perceptron is online instead, then the statistical complexity improve from $O((1/\gamma)^2)$ to $O(1/\gamma \ln(1/\gamma))$ which is slightly less than quadratic. The computational improvement is due to the quantum search while the statistical improvement is provided by our choice of using a version space based strategy, leading to the name 'hybrid QP'. Theorem 4 shows that our algorithm is particularly well-suited for large-scale data sets and small margins.

---

**Algorithm 5:** HYBRID QUANTUM PERCEPTRON

**Input:** data $(x_i, y_i)_{1 \leq i \leq N}$ ;          // training set
**Input:** $\{w_1, \ldots, w_K\} \sim \mathcal{N}(0, \mathbb{1})$ ;     // hyperplanes
**for** $i \in \{1, \ldots, K\}$ ;        // for all hyperplanes...
**do**
  $b \leftarrow 1$ **for**
  $j \in \{1, \ldots, \left\lceil \log_{3/4}\left(1 - \left(1 - \frac{\epsilon}{2}\right)^{\frac{1}{K-1}}\right)\right\rceil\}$ ;
  // increase QSEARCH success probability
  **do**
    $m \leftarrow \text{QSEARCH}(\{(x_k, y_k)\}_k)$ ;   // searching for a point $x_m$ misclassified by $w_i$
    **if** $y_m w_i^T x_m \leq 0$ ;        // if one is found...
    **then**
      $b \leftarrow 0$ ; // ...then the current hyperplane isn't a good one
  **if** $b = 1$ ;      // if no miclassified point found...
  **then**
    **return** $w_i$ ;        // ...then return the current hyperplane
**return** $w_1$

---

## 4.2 GENERALIZATION

In the classical setting, mistake bounds for the Perceptron algorithm can be used to derive generalization bounds [Cesa-Bianchi et al., 2004, Mohri and Rostamizadeh, 2013]. This question was not addressed in Wiebe et al. [2016]. As we have seen above, the HYBRID QUANTUM PERCEPTRON provides an improvement on the statistical efficiency of the perceptron ($O(1/\gamma)$ instead of $O(1/\gamma^2)$). We show here that this may yield better generalization guarantees.

We have a training set $S = \{z_1, \ldots, z_n\}$ with $z_i = (x_i, y_i)$. We assume that $z_i$ are independently sampled from an unknown distribution $\mathcal{D}$. We recall that the risk is defined by

$$
R(h) = \mathbb{E}_{z \sim \mathcal{D}} \left(\mathbb{1}\{h(x) \neq y\}\right) ,
$$

where $h$ is a hypothesis in a hypothesis set $\mathcal{H}$.

**Theorem 5 .** *Assume that the data is linearly separable. Let $h_S$ be the hypothesis returned by the* HYBRID QUANTUM PERCEPTRON *algorithm after training over a sample $S$ of size $N$ drawn according to some distribution $\mathcal{D}$. Then, the expected error of $h_S$ is bounded as follows:*

$$
\mathbb{E}_{S \sim \mathcal{D}^N} \left(R(h_S)\right) \leq \sqrt{\frac{\pi}{2}} \frac{\log 1/\epsilon}{N+1} \mathbb{E}_{S \sim \mathcal{D}^{N+1}} \left(\frac{1}{\gamma_S}\right) .
$$

*Proof.* See supplementary materials. ☐

The bound obtained in the classical online setting is equal to $\frac{1}{N+1} \mathbb{E}_{S \sim \mathcal{D}^{N+1}} \left(\min(M(S), 1/\gamma_S^2)\right)$, where $M(S)$ is the number of updates made by the algorithm after training over $S$ [Mohri et al., 2018, Theorem 8.9]. Theorem 5 shows that

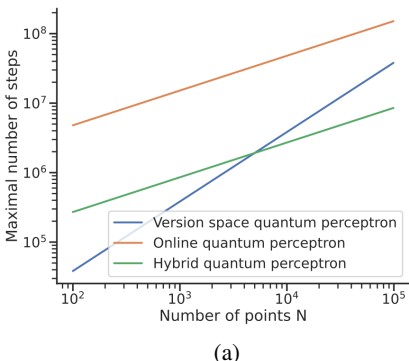
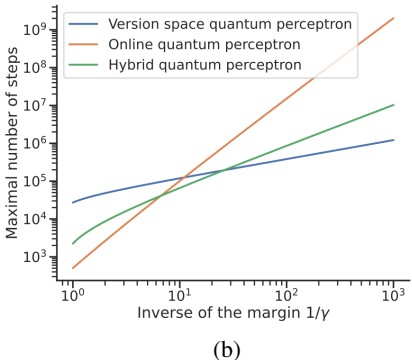

|        (a)        |        (b)        |

Figure 1: Complexity bounds over the number of operations for quantum perceptrons. The curves asymptotically follow the complexities summarized in Table 1. Subplot (a) shows the bounds in function of the number of points $N$ with a margin $\gamma = 0.01$. Subplot (b) shows the bounds in function of the inverse of the margin $\gamma$ with a number of points $N = 1000$.

HYBRID QUANTUM PERCEPTRON can give considerable improvement in generalization over the classical online perceptron algorithm. However, the guarantee given is not a high probability bound, since it holds only for the expected error of the hypothesis returned by the algorithm.

## 5 NUMERICAL EXPERIMENTS

In this section, we illustrate empirically the theoretical performance guarantees introduced in the previous section. Then , we discuss the effect of quantum noise which is one of the major issue of near-term quantum algorithms. The simulations presented here come from a classical computer simulating a quantum algorithm.[1]

### 5.1 COMPUTATIONAL-STATISTICAL TRADE-OFF

We run experiments with the three quantum perceptron models studied in this paper and compare the number of steps required for these algorithms when varying the number of data samples $N$ and the margin $\gamma$. Figure 1 shows the maximal number of steps; namely the complexities taking into account the constants. The slope of the curves gives an indication of the complexity in terms of $N$ or $1/\gamma$, while the intercept provides a good indication of the impact of the constant factors on it. The slope of the curve of HYBRID QUANTUM PERCEPTRON is lower than the one of the VERSION SPACE QUANTUM PERCEPTRON when $\gamma$ is fixed and $N$ varies and also lower than the slope of ONLINE QUANTUM PERCEPTRON when $N$ is fixed and $\gamma$ varies. This confirms that our algorithm has a lower computational

complexity and also a better statistical efficiency.

It is also interesting to compare the behavior of these quantum perceptron algorithms with respect to the number of operations made by the classical online perceptron. We apply the three quantum perceptron algorithms on the Iris dataset and on a simulated dataset (called Hard). Iris is a simple dataset for which the classical perceptron will converge very quickly. The Hard dataset, however, is specifically build to force the classical perceptron algorithm to perform a large number of updates.

**Definition 1** (Hard dataset)**.** The Hard dataset inspired from Mohri et al. [2018, Exercice 8.1] is composed of a sample $S_H(N) = \{(x_1, y_1), \ldots, (x_N, y_N) \in (\mathbb{R}^N \times \{0, 1\})^N$ of size $N$ and dimension $N$ such that

$$\forall i, j \in [N]^2, \ (x_i)_j = (-1)^{i+1} \mathbb{1}\{j = i\} \text{ and } y_i = (-1)^i .$$

Figure 2 shows the ratio between the number of operations of each quantum perceptron algorithm and the number of steps of the classical perceptron during the learning phase. On the Iris dataset, the three quantum perceptrons behave similarly and are about four times slower than the classical perceptron. This is expected since the problem is easy to solve. For the Hard dataset, however, all the quantum perceptron algorithms shows an improvement over the classical one. Interestingly, HYBRID QUANTUM PERCEPTRON is the one that performs the best, since it achieves a good trade-off between computational and statistical complexities.

### 5.2 QUANTUM NOISE

Most of the existing quantum devices are subject to quantum noise. Dealing with noise in quantum computation is nowadays an important and challenging problem. Although a rigorous analysis goes beyond the scope of this work,

---

[1]The code to reproduce our experiments is available in a GitHub repository: `https://github.com/mroget/Quantum-perceptron-models`.

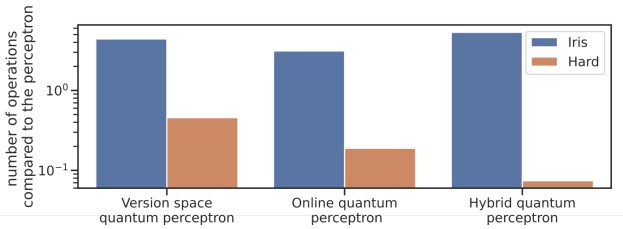

Figure 2: Ratio between the number of operations of quantum perceptron and classical perceptron.

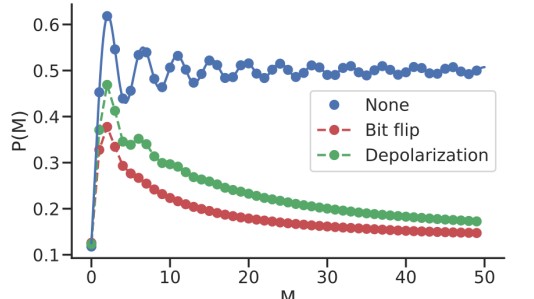

Figure 3: $P(M)$ for different noise models.

here we shortly illustrate how noise may affect the quantum perceptron computational task. All quantum algorithms presented in this paper are based on the assumption that the quantum search finds a searched element with probability at least $1/4$. As a reminder (see Section 2), the quantum search is designed by performing $m$ steps of the Grover's algorithm while $m$ is drawn uniformly between 0 and $M$. We can compute the probability of success of the quantum search with respect to $M$. Let us call this probability $P(M)$. As seen in Section 2, it holds that

$$P(M) = \frac{1}{2} \left( \frac{\sin(4M\theta_a)}{2M\sin(2\theta_a)} \right) .$$

Here, $\theta_a$ depends only on the proportion of searched elements. Figure 3 shows three curves. Each one is the plot of $P(M)$ for one searched element with a specific quantum noise model. The blue one does not account noise while the other two curves have, respectively, bit-flip, and depolarization noise [Wang and Krstic, 2020]. The first class of noise coincides with a unitary random flip, meaning that the computational state flips from $|1\rangle$ to $|0\rangle$ or vice versa. The second kind of error can be seen as a completely positive trace-preserving map from the quantum state onto a linear combination of itself and a general maximally mixed state. As we can see, the success probability in a fault-free environment converges towards $1/2$, thus for large enough $M$, $P(M)$ can be always greater than $1/4$, as explained in the section Preliminaries. However in a faulty-environment, the success probability decreases rapidly and do not tend to a non vanishing constant, making harder to recover a $P(M)$ greater than $1/4$. Moreover the quantum noise strictly depends on the quantum circuit design (in concrete how errors may propagate), making this choice crucial to build a fault-tolerant quantum perceptron. This decreasing is the result of making too many iterations, thus accumulating noise. On the other hand, the probability starts by increasing because the quantum search is working. The peak of probability represents the best trade-off between the increase of the probability of success and the increase of the quantum noise.

## 6 DISCUSSION

In this work, by classical perceptron we mean the standard online perceptron. There is, to our knowledge, no mention in the classical ML literature to classical version space perceptron. The Quantum Perceptron algorithm we propose has a quadratic improvement in $N$ and $\gamma$ over the well-known classical online perceptron. Similarly, a quartic speed-up is used in Wiebe et al. [2016] to describe the improvement over $\gamma$ they obtained with their quantum version space perceptron. It is worth noting that, although it is not known in the literature, a classical version space perceptron should have a complexity bound inversely proportional to the margin $\gamma$. The quadratic improvement over the margin is not provided by the Grover's search algorithm but by the version space approach. Usually the version space approach scales linearly with the number of examples $N$. The quadratic improvement in $N$ is, however, obtained by our quantum perceptron using a quantum search over the training set. When adopting a version space approach, the perceptron problem is transformed into a search problem over the generated hyperplanes. Our results show that applying a quantum search over the training set and not over the hyperplanes in this situation provides new insights for the design of computationally and statistically efficient perceptron models.

To our knowledge, our Theorem 5 is the first result showing that the version space perceptron (classic or quantum) can have a better generalization than the online perceptron algorithm. There are no results concerning the expected risk of previous quantum perceptron algorithms. We expect that the expected risk bound of ONLINE QUANTUM PERCEPTRON is of the same order than the classical online perceptron, since this algorithm does not improve the mistake bound. For VERSION SPACE QUANTUM PERCEPTRON, it is not clear whether the improvement on the scaling of the algorithm with respect to the margin could yield even better generalization guarantees. The factor $1/\gamma$ in the expected risk bound of HYBRID QUANTUM PERCEPTRON is related to the number of the randomly generated linear separators (see the proof of Theorem 5). The version space quantum perceptron has the same number of separators than our algorithm. So, using the same line of proof as for Theorem 5 will not necessarily result in an improved bound.

In this paper we only considered linear classification. In

the classical case, kernel methods provide a powerful tool for generalizing linear classifiers to nonlinear settings [Schölkopf et al., 2002]. With appropriate nonlinear features, linear models can be used to approximate a nonlinear function. Kernel methods allow the construction of these nonlinear features. There are interesting links between kernel methods and quantum computing [Havlíček et al., 2019, Schuld and Killoran, 2019]. Indeed, the process of encoding inputs in a quantum state can be interpreted as a nonlinear feature map that maps data to a quantum Hilbert space. So, the quantum encoding of classical data can be seen as a way to construct nonlinear quantum features. Different quantum encodings were proposed and the corresponding kernels were given. Nonlinear extensions of our work can be carried out by the classical-to-quantum data encoding scheme.

# 7 CONCLUSION

In this paper, we proposed a hybrid quantum perceptron algorithm that goes beyond the ideas of existing quantum perceptron algorithms. This allowed us to obtain a quadratic improvement over the computational complexity and the statistical efficiency compared to the classical online perceptron. We performed numerical experiments to support our theoretical findings. In the future, it would be valuable to study noise-robust models for quantum perceptron.

### Acknowledgements

We thank L. Ralaivola for useful discussions. This work has been funded by the French National Research Agency (ANR) project QuantML (grant number ANR-19-CE23-0011) and the INS2I CNRS project QuAlgo.

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
