# OpenReview forum: "Quantum Perceptron Revisited: Computational-Statistical Tradeoffs"
_auai.org/UAI/2022/Conference — UAI 2022 Oral_

### Official Review · Reviewer_m2fc · 2022-03-28

**Q2(1) Originality/Novelty:** 3
**Q2(2) Significance/Impact:** 3
**Q2(3) Correctness/Technical Quality:** 4
**Q2(6) Clarity Of Writing:** 4
**Q6 Overall Score:** 8
**Q8 Confidence In Your Score:** 3

**Q1 Summary And Contributions:**

The paper presents a novel quantum perceptron algorithm that  combines aspects of two existing quantum algorithms, the Online Quantum Perceptron(OQP) and the Version Space Quantum Perceptron(VSQP). The authors prove the algorithm has complexity $O(\sqrt{N}/\gamma)$; $N$ is the sample size and $\gamma$ is the margin; this improves the computational-statistical tradeoff of OQP and VSQP which scale like $O(N/\sqrt{\gamma})$ and $O(\sqrt{N}/\gamma^2)$ respectively. A generalisation bound is proved.

**Q2 Assessment Of The Paper:**

More detailed information regarding each of these aspects is given below:

**Q2(4) Quality Of Experiments (Optional):**

4: Excellent: The experimental evaluation is comprehensive and the results are compelling.

**Q2(5) Reproducibility:**

4: Excellent: Key resources (e.g., proofs, code, data) are available and key details (e.g., proof sketches, experimental setup) are comprehensively described for competent researchers to confidently and easily reproduce the main results.

**Q3 Main Strengths:**

The paper is very well written. It provides a novel algorithm, based on a new idea, which achieves a better statistical-computational tradeoff than the existing algorithms. The proofs seem correct.  Code is provided.

**Q4 Main Weakness:**

I did not identify any important weaknesses.

**Q5 Detailed Comments To The Authors:**

*page 3, col 2, l-11: “are nothing both” should possibly be “are nothing but”


**Q7 Justification For Your Score:**

This is a strong paper. As far as I understand it is technically correct and novel, proposing a new quantum algorithm for binary classification that achieves a much better tradeoff between dependence on the margin and dependence on the sample size. I am not an expert in the area, but as far as I understand the paper has potential for impact in the area.

**Q9 Complying With Reviewing Instructions:**

1: Yes.

---

### Official Review · Reviewer_NhxC · 2022-04-03

**Q2(1) Originality/Novelty:** 2
**Q2(2) Significance/Impact:** 2
**Q2(3) Correctness/Technical Quality:** 1
**Q2(6) Clarity Of Writing:** 3
**Q6 Overall Score:** 7
**Q8 Confidence In Your Score:** 4

**Q1 Summary And Contributions:**

The paper combines two quantum perception algorithms: the online quantum perceptron and the version space quantum perceptron.
The algorithm that is obtained is a hybrid of the two and present parts of the advantages of both in terms of complexity with respect
to the number of the examples and the margin. Moreover, the paper presents a results regarding the expected error
that is not available for the other two algorithms, considerations on the implementation and an experimental evaluation.


**Q2 Assessment Of The Paper:**

More detailed information regarding each of these aspects is given below:

**Q2(4) Quality Of Experiments (Optional):**

4: Excellent: The experimental evaluation is comprehensive and the results are compelling.

**Q2(5) Reproducibility:**

1: Poor: Key details (e.g., proof sketches, experimental setup) are incomplete/unclear, or key resources (e.g., proofs, code, data) are unavailable.

**Q3 Main Strengths:**

The paper presents a simple hybridization of two quantum perceptron algorithm that combines parts of the advantages of both.

The paper provides generalization bounds for the new algorithm that are not available for the two hybridized algorithms.

The paper provides an experimental evaluation showing where the algorithm is advantageous.

The paper provides a discussion of the effects of quantum noise on the algorithm.

**Q4 Main Weakness:**

According to the experimental evaluation, the advantages in terms of the margin occur only in a short interval: for very large margin the online algorithm is better, for very small the version space one is better

The reproducibility of the paper is poor as the proofs of the two main theorems (complexity and generalization bound) are
missing key details and passages. Moreover, the proof of one theorem seems to have a mistake.

**Q5 Detailed Comments To The Authors:**

The idea behind the algorithm is relatively simple but seems relatively effective, with better complexity than the online version in terms of number of examples and inverse of the margin and better complexity than the version space in terms of the number of examples.

The impact is fair, it will mainly influence the community working on quantum algorithms for machine learning.

The soundness of the paper is difficult to judge, as the proofs of the main theorems in the appendix lack many passages that are difficult or impossible to reconstruct, see the punctual comments below.
This makes the reproducibility poor.  Moreover, the use of the big O value for K in the proof of Theorem 4 for computing the bound on the probability that a randomly sampled hyperplane exists that separates the data seems not legit.

The experiments are excellent, clearly showing where the algorithm is useful and where it is not.

The paper is overall clear but there are a few points to improve, see the punctual comments below.

Punctual comments:
Theorem 1: say what  \gamma_S is

Page 3: In the second formula for \ket{\psi}, the upper bound of the summation is 2^n-1, not n-1

"Quantum circuits are nothing both reversible logical"->"Quantum circuits are nothing but reversible logical"

"computational basis state \ket{v} \in C^n"->"computational basis state \ket{v} \in C^{2n}"

Page 4: "The idea here that comes from Wiebe et al. [2016] is simply": the idea of sampling the number of steps of Grover does not seem to come from that paper. Moreover, the value of M in alg 2 is not the one shown below: in the one show below #M appears, while in alg 2 it is replaced by 1, how do you justify this?
In the formula for the bound on M, the term \leq O(\sqrt{N}) should be replaced by \in  O(\sqrt{N})

In the proof of Theorem 4, I cannot fill in the steps to go from the value you assign to K to O(ln(1/espilon)/\gamma).
In the computation of the probability that a separating w exists, 1-\sqrt{2/\pi}\gamma must be the probability that a
randomly sampled hyperplane does not separate the data: where does this formula come from?
I cannot fill in the steps to go from 1-(1-\sqrt{2/\pi}\gamma)^K to 1-\espilon/2. Only if I use the O value for K I get the results but using an O for K does not seem to be legit since you find a smaller value for (1-\sqrt{2/\pi}\gamma)^K.
I cannot fill it the steps to from the formula for K_2 to O(ln(1/e))

In the proof of Theorem 5, if N\leq K, than K/N is larger than 1, so the bound is vacuous.
I cannot fill it in the details on how you get the final formula from Lemma 1.

**Q7 Justification For Your Score:**

Interesting idea but the paper lacks the details to evaluate its technical soundness and the proof of one theorem has a major flaw.

---------------
The authors' response clarified my doubts, the paper can be accepted.

**Q9 Complying With Reviewing Instructions:**

1: Yes.

---

### Official Review · Reviewer_egee · 2022-04-12

**Q2(1) Originality/Novelty:** 3
**Q2(2) Significance/Impact:** 3
**Q2(3) Correctness/Technical Quality:** 3
**Q2(6) Clarity Of Writing:** 4
**Q6 Overall Score:** 7
**Q8 Confidence In Your Score:** 1

**Q1 Summary And Contributions:**

The paper proposes a hybrid quantum-classical perceptron algorithm that achieves improvements over the classical perceptron, in terms of both complexity and generalization. The claims are further evidenced by numerical results.

**Q2 Assessment Of The Paper:**

More detailed information regarding each of these aspects is given below:

**Q2(4) Quality Of Experiments (Optional):**

3: Good: The experimental evaluation is adequate, and the results convincingly support the main claims.

**Q2(5) Reproducibility:**

3: Good: Key resources (e.g., proofs, code, data) are available and key details (e.g., proofs, experimental setup) are sufficiently well-described for competent researchers to confidently reproduce the main results.

**Q3 Main Strengths:**

1. The paper obtains state-of-the-art results in terms of quantum/hybrid perceptron.
2. The paper is well written.

**Q4 Main Weakness:**

I am afraid I do not have sufficient expertise to weigh in on this.

**Q5 Detailed Comments To The Authors:**

I think the paper is cleanly presented. I wish I had more backgrounds on the subject to understand the technical contributions.

**Q7 Justification For Your Score:**

Due to my lack of expertise on quantum computing (I did not bid this article), I am giving this score based on my educated guess and the fact that the paper is cleanly written.

**Q9 Complying With Reviewing Instructions:**

1: Yes.

---

### Decision · Program_Chairs · 2022-05-15

**Decision:**

Accept (Oral)

**Comment:**

Meta Review: The reviewers and myself agree that this is an interesting quantum take on a classical problem --- generalization of the perceptron algorithm in the presence of margin. The hybrid classical-quantum algorithm (using Grover search) proposed has a quadratic improvement over the classical algorithm. The proof techniques are interesting, and broadly accessible and interesting for the UAI community.